# Microbial Communities in Gynecological Cancers and Their Association with Tumor Somatic Variation

**DOI:** 10.3390/cancers15133316

**Published:** 2023-06-23

**Authors:** Jesus Gonzalez-Bosquet, Megan E. McDonald, David P. Bender, Brian J. Smith, Kimberly K. Leslie, Michael J. Goodheart, Eric J. Devor

**Affiliations:** 1Division of Gynecologic Oncology, Department of Obstetrics and Gynecology, University of Iowa Hospitals and Clinics, Iowa City, IA 52242, USA; megan-mcdonald@uiowa.edu (M.E.M.); david-bender@uiowa.edu (D.P.B.); michael-goodheart@uiowa.edu (M.J.G.); eric-devor@uiowa.edu (E.J.D.); 2Holden Comprehensive Cancer Center, University of Iowa Hospitals and Clinics, Iowa City, IA 52242, USA; 3Department of Biostatistics, University of Iowa, Iowa City, IA 52242, USA; brian-j-smith@uiowa.edu; 4Division of Molecular Medicine, Department of Internal Medicine and Obstetrics and Gynecology, The University of New Mexico Comprehensive Cancer Center, Albuquerque, NM 87131, USA; kkleslie@salud.unm.edu

**Keywords:** microbial communities, metagenomics, high grade serous ovarian cancer, endometrioid endometrial cancer, bacterial, single nucleotide variation

## Abstract

**Simple Summary:**

Bacteria are responsible for a wide range of human diseases including cancer. In our study we identified changes in the microbial communities of the female upper genital tract correlated with genomic variation in genes seen in gynecological cancers, endometrial and ovarian. This supports our hypothesis that differences in bacterial communities in the upper genital tract epithelium may lead to selection of specific genomic variation at the cellular level of these tissues that may predispose to their malignant transformation. Pathway analyses including correlated genes with changing microbiome may help understand how changing bacterial landscapes could lead to these cancers.

**Abstract:**

There are strong correlations between the microbiome and human disease, including cancer. However, very little is known about potential mechanisms associated with malignant transformation in microbiome-associated gynecological cancer, except for HPV-induced cervical cancer. Our hypothesis is that differences in bacterial communities in upper genital tract epithelium may lead to selection of specific genomic variation at the cellular level of these tissues that may predispose to their malignant transformation. We first assessed differences in the taxonomic composition of microbial communities and genomic variation between gynecologic cancers and normal samples. Then, we performed a correlation analysis to assess whether differences in microbial communities selected for specific single nucleotide variation (SNV) between normal and gynecological cancers. We validated these results in independent datasets. This is a retrospective nested case-control study that used clinical and genomic information to perform all analyses. Our present study confirms a changing landscape in microbial communities as we progress into the upper genital tract, with more diversity in lower levels of the tract. Some of the different genomic variations between cancer and controls strongly correlated with the changing microbial communities. Pathway analyses including these correlated genes may help understand the basis for how changing bacterial landscapes may lead to these cancers. However, one of the most important implications of our findings is the possibility of cancer prevention in women at risk by detecting altered bacterial communities in the upper genital tract epithelium.

## 1. Introduction

There are strong correlations between the microbiome (microbial community occupying the human body) and human disease, such as metabolic disorders [1], infectious diseases [2], digestive system diseases [3], and cancers [4]. An estimated 15–20% of cancers worldwide are linked to viral, parasitic, or bacterial infections [4]. Some of the most known cancer-associated infectious agents are viruses, such as hepatitis B virus (HBV), human papillomavirus (HPV), Epstein–Barr virus (EBV), and human immunodeficiency virus (HIV), which integrate their genetic material into the human genome [5]. The integration of HPV genetic material is one of the best known mechanisms of malignant transformation, leading to the production of E6 and E7 oncoproteins that bind and inactivate p53 and the retinoblastoma family of tumor suppressor proteins, bypassing cell cycle controls [6,7]. *H. pylori* infection is associated with bacterial-induced malignant transformation [8]: a persistent *H. pylori* infection over decades provides a chronic inflammatory state capable of inducing cell damage and mutagenesis, leading to gastric metaplasia and, in selected individuals, clones of neoplastic clones under the pressure of DNA replication. This association of changes in the bacterial milieu and the resultant altered cellular microenvironment may favor certain alterations in specific cancers [8].

The Human Microbiome Project Consortium [9,10], has studied the human microbiome, including the normal microbiome of the vaginal environment. Additionally, there are a few studies about the microbiome in the upper genital tract and its association with gynecological (uterine and ovarian) cancers [11,12]. However, very little is known about potential mechanisms associated with malignant transformation in microbiome-associated gynecological cancer, except for HPV-induced cervical cancer [6,7]. Additionally, all cancers harbor a myriad of genomic variations, either in the form of single nucleotide variation (SNV), copy number variation (CNV) or structural variation. The pattern of these variations are characteristics of each neoplasia [13].

Our hypothesis is that differences in bacterial communities in the upper genital tract epithelium may lead to selection of specific genomic variation at the cellular level of these tissues that may predispose to their malignant transformation. These microbiome differences and genomic selection may also influence the heterogeneity of these cancers and their potential outcomes. To test this hypothesis, we first assessed differences in the taxonomic composition of microbial communities between endometrial cancer and endometrial normal tissues, and between tubal and ovarian cancer tissues. Then we assessed SNV differences between normal endometrium and endometrial cancer, and between tubal and ovarian cancer tissues. Finally, we performed a correlation analysis to assess whether differences in microbial communities selected for specific SNV between normal and gynecological cancers. We validated these results in independent datasets, the Cancer Genome Atlas (TCGA), and Gene Expression Omnibus (GEO) databases.

## 2. Materials and Methods

This is a retrospective nested case-control study that used clinical and genomic information to create and compare taxonomic composition of microbial communities from high-grade serous ovarian cancer (HGSC) and endometrioid endometrial cancers (EEC) and their correlation with genomic variation in the same samples. Due to increasing evidence of the genesis of HGSC in the fallopian tube, we used nonmalignant fallopian tube samples as a normal control for HGSC [14]. We then validated the results from this analysis with the TCGA HGSC and EEC databases.

### 2.1. Sample Acquisition

HGSC and EEC tissue samples, pathological classification, and clinical outcomes, were collected during the surgical treatment of patients with HGSC and were stored at the Department of Obstetrics and Gynecology Gynecologic Oncology Bank (IRB ID#201809807), which is part of the Women’s Health Tissue Repository (WHTR, IRB ID#200209010) [15]. Tumor samples were collected, reviewed by a board-certified pathologist, flash-frozen, and then the diagnosis was confirmed in paraffin at the time of initial surgery. All experimental protocols were approved by the UI Biomedical IRB-01. All samples were collected under informed consent in accordance with the University of Iowa (UI) IRB guidelines. As controls, we chose fallopian tubes as this is the most likely origin of HGSC based in the most recent knowledge in ovarian cancer pathophysiology [16]. We collected 20 normal fallopian tube samples in coordination with the UI Tissue Procurement Core Facility, with independent consent (IRB, ID#201202714). The tissue extracted from the tubes came from the junction of the ampullary and fimbriated end of fallopian tubes. The volunteers that donated the tubal samples for this study had no family or personal history of ovarian cancers, or other cancers, and were scheduled to undergo elective salpingectomy for sterilization. Endometrial control samples were obtained from a publicly available source, GEO. These were 36 samples from healthy patients who underwent a fertility study wherein an endometrial biopsy was taken, frozen, and RNA was extracted and processed (GSE180485) [17].

Of the 412 initial HGSC candidates for the study, 193 patients were identified to have frozen tissue, and of those, 112 patients had enough tissue with RNA yield and quality for analysis [18]. Similarly, of the initial 155 EEC patients identified in the Biobank search, 127 had the tumor initially frozen, and 62 patients still had the primary tumor with sufficient RNA yield and quality for analysis [19]. From the 20 original normal fallopian tube samples, 12 had sufficient RNA yield and quality for analysis [18] (Figure 1). Selected specimens for analysis had less than 30% necrosis.

### 2.2. DNA and RNA Purification and Sequencing

Genomic DNA (gDNA) and total cellular RNA were purified from flash frozen tumor (ovarian and endometrial) and normal tissues, as described previously [20]. Only RNAs with an RNA integrity number (RIN) [21] greater than or equal to 7.0 were selected for RNA sequencing. Each qualifying tumor was fragmented, converted to cDNA and ligated to bar-coded sequencing adaptors using Illumina TriSeq stranded total RNA library preparation (Illumina, San Diego, CA, USA). The yield and purity of Genomic DNAs were assessed on a NanoDrop Model 2000 spectrophotometer and by using horizontal agarose gel electrophoresis.

Indexed libraries were confirmed on the Agilent Model 2100 bioanalyzer and libraries were then combined into pools for sequencing. Sequencing for both RNA and DNA was then conducted on the Illumina HiSeq 4000 genome sequencing platform using 150 bp paired-end SBS chemistry. All library preparation and sequencing were performed in the Genome Facility of the University of Iowa Institute of Human Genetics (IIHG). Quality control (QC) of RNA sequencing experiments (RNA-seq) and whole genome sequencing (WES) were performed to minimalize technical biases.

### 2.3. Taxonomic Composition of Microbial Communities

Taxonomic composition of microbial communities is performed on a large set of genomes as opposed to just one reference genome [22]. In the present study, we used the sequence analysis of the 16S rRNA gene-based amplicon to determine structures of microbial communities [23]. For that purpose, we used the tool *VITCOMIC2* (VIsualization tool for Taxonomic COmpositions of MIcrobial Community) [22]. *VITCOMIC2* compares 16S rRNA gene sequences from selected samples with reference 16S rRNA gene sequences and identifies the nearest relative of each submitted sequence. The reference 16S rRNA gene database used for this study was obtained from the Ribosomal Database Project (RDP), including 1,345,732 16S rRNA gene sequences of Bacteria and Archaea (release 11, update 2) [24]. The RDP was modified to obtain a curated database with 28,977 high-quality 16S rRNA gene sequences [22].

We used VITCOMIC2 for its simplicity, speed, and accuracy [25]. However, the main reason was the versatility that allows us to use *fastq* files from both DNA-seq and RNA-seq experiments, and still map the sequences to the reference sequences. For the *VITCOMIC2* analysis, we first obtained *fastq* files from RNA-seq experiments of all samples. Then, *fastq* files from sequencing were pre-processed with *fastp* for quality control, adapter trimming, quality filtering, and per-read quality pruning [26]: 3′ tail was cut, limiting the analysis of transcripts of at least 90 bp, and 3′ polyX trimming. Then, *seqKit* (version 2.1.0) was applied to convert *fastq* to *fasta* format (fq2fa) [27]. Finally, *MAPseq* (version 2.0.1 alpha) was used to map sequences against hierarchically clustered and annotated reference 16S rRNA sequences [28]. Additionally, *MAPseq* provides a large, curated reference of full-length rRNA genes, pre-clustered into operational taxonomic unit (OUT) at different identity thresholds, and pre-classified to taxonomic categories based on the NCBI taxonomy and the All-species Living Tree Project dataset [29]. We used hits with an identity ≥ 94% and an alignment length ≥75 bp, as recommended by the authors [22]. *Phyloseq* (version 1.0.3) was used for the representation and analysis of microbiome census data from these hits [30]. We took the transcript counts for all indexes and used the *DESeq2* package (version 1.40.1) to import, normalize, and log2-transform the data for further analysis [31,32].

### 2.4. Somatic Single Nucleotide Variation (SNV) Analysis

*Subread* (version 2.0.3) was used to align the RNA-seq reads to the human reference genome (version hg38) and to create BAM files. Then, we used these BAM files to create Variant Call Format (VCF) using the software *samtools* (version 1.13) [33] and *VarScan* (version 2.4.3) [34], as recommended by best practices of genome sequencing [35]. Filtering parameters for *VarScan* were set to frequency >0.02, *p*-value < 0.05, minimum coverage of 8, minimum quality score of 15. Then, VCF files were transformed into MAF files, using the *vcf2maf* utility (version 2.0) which will annotate VCFs, prioritize transcripts, and generate an MAF. *Vcf2maf* utility uses the Ensembl Variant Effect Predictor (VEP) to determine the effect of variants (SNPs, insertions, deletions, CNVs or structural variants) on genes, transcripts, and protein sequences, as well as regulatory regions [36]. We removed all inconsequential variants and retained only those present upstream of transcripts, coding regions, regulatory regions, and non-coding RNA, that have downstream consequences (i.e., missense, frameshift, stop gained or lost) [37]. For analysis, we constructed a specific table with resulting SNVs from all samples.

### 2.5. Statistical Analysis

#### 2.5.1. Comparison of Bacterial 16S rRNA Counts between Control and Cancer Samples

Comparisons between 16S rRNA log2-transformed, normalized counts between control and cancer samples, determined as previously described, were performed initially with a univariate regression analysis, adjusted with false discovery rate (FDR). Then, transformed counts initially significant in the univariate analysis were introduced in a multivariate model to assess for independence or prediction (logistic or lasso modelling). Performance of prediction models were measured by the area under the curve (AUC) and their respective 95% confidence intervals (CI). A *p*-value < 0.05 was considered significant. Most statistical analyses were performed using R statistical package for statistical computing and graphics (www.r-project.org, accessed on 5 August 2023) as background, using Bioconductor packages as open-source software for bioinformatics (bioconductor.org, accessed on 5 August 2023).

#### 2.5.2. Comparison of SNVs between Control and Cancer Samples and Survival Analyses

Comparison of SNVs between control and cancer samples were performed with the *maftools* R package (version 2.16.0) [37]. Differences between both groups were detected using function *mafComapre*, which performs Fisher’s test on all genes between two groups to detect genes with different SNVs, adjusted with FDR. A *p*-value < 0.05 was considered significant. For further analyses (multivariate modelling and correlation with 16S rRNA count) the SNV counts data table was downloaded into *DESeq2*, normalized, and log2 transformed.

Survival analysis of genes with differential SNV analysis was also performed with *maftools*, using Kaplan–Meier graphics and computing differences on survival between mutated and non-mutated samples using Cox proportional hazard ratio (HR).

#### 2.5.3. Correlations between Significant 16S rRNA Counts and SNVs

Correlations between significant 16S rRNA counts and significant SNVs in comparisons between control and cancer tissues were performed using Spearman’s rank correlation test, as the expression between these genomic elements is not completely independent from one another. Statistical significance was assessed with *p*-value and FDR correction for multiple comparisons [38].

#### 2.5.4. Validation of Analysis between Control and Cancer Samples in TCGA Dataset

We used the TCGA HGSC and EEC databases to validate 16S rRNA expression. We downloaded BAM files from the TCGA website (Genomic Data Commons—GDC—Data Portal, dbGaP# 29868). BAM files were transformed to *fastq* format with *bedtools* (version 2.30.0), a suite of tools for genomic analysis [39]. Then *fastq* files were processed as previously to obtain normalized, log2-transformed 16S rRNA from the TCGA HGSC and EEC databases. Some of TCGA older RNAseq analyses for the EEC dataset were resulted in 50 mers (nucleotide sequence of 50 bp). That did not conform to the specifications of the *VITCOMIC2* process for all other specimens: identity ≥ 94% and an alignment length ≥75 bp. For these older RNAseq 50 mer samples we used an alignment length ≥45 bp.

Additionally, *fastq* files were processed to get SNV frequencies from both types of cancers. For validation of previous results, only 16S rRNA counts and SNV that were considered significant between both groups in previous comparisons were assessed in TCGA analysis. Because TCGA lacks suitable controls to compare with HGSC and EEC samples, we used the same controls used in the previous analysis.

#### 2.5.5. Power Calculation

To achieve 80% power to find 16S rRNA counts with a mean difference of 1 in expression (log2-transformed) between classes, and 0.005 Type 1 error, assuming the 50th percentile of the variance distribution, we will need 13 samples per group for the analysis. If we assume the 75th percentile of the variance distribution, we would need 23 samples per group.

### 2.6. Pathway Enrichment Analysis

We used *clusterProfiler* R package (version 4.8.1), a platform that uses *WikiPathways* as a curated knowledge-based database [40,41] to identify over-represented pathways among the significant list of genes with SNV correlated with significant 16S rRNA. The FDR-adjusted *p*-value represents the probability that a particular gene from the list of resulting genes is located in a specific pathway by chance, taking into account all genes present in all pathways.

## 3. Results

### 3.1. Comparison of Bacterial 16S rRNA Counts between Control and Cancer Samples

After extraction of 16S rRNA expression from HGSC and control samples, we identified 801 taxa for all 124 samples (Figure 2A). In the univariate logistic regression analysis, 13 of these taxa were statistically different between HGSC and normal tubes (*p* < 0.05) (Figure 2B). In the endometrial dataset (EEC and control), after the 16S rRNA expression analysis, we found 655 unique taxa for the 98 samples (Figure 2C). In the univariate analysis, 112 of them have statistically different counts of 16S rRNA (Figure 2D).

16S rRNA expression is statistically different between HGSC and normal fallopian tubes (*n* = 13, Figure 3A) were introduced in multivariate models: (1) the multivariate logistic regression model identified those taxa independently significant, genera *Leclercia* and unclassified *Desulfobulbaceae* (Figure 3B); (2) the multivariate lasso regression model determined those bacteria that would predict the presence of HGSC (Figure 3C). This prediction model had a performance of 0.97 measured by the AUC, with a 95% CI of 0.94, 1.00.

The multivariate lasso regression analysis between EEC and control endometrial samples that included all 16S rRNA counts significant in the univariate analysis (*n* = 112), resulted in four 16S sRNAs that predicted EEC with an impressive performance, AUC of 1.00: *Desulfobacter*, *Desulfomicrobium*, *Parabacteroides*, and *Proteus* (Appendix A). In each case, EEC had reduced 16S rRNA gene expression (OR < 1).

### 3.2. Comparison of SNVs between Control and Cancer Samples and Survival Analyses

In the HGSC dataset, including 112 cases, there were a total of 748,433 variants upstream of transcripts, coding regions, regulatory regions, and non-coding RNA that had downstream consequences: frame-shift deletion/insertion, in-frame deletion/insertion, missense, non-sense or non-stop mutations, and splice site. Appendix A summarized these variations processed with *maftools*. The normal fallopian tube set (with 12 controls) had 43,961 SNVs (Appendix A). All these variants were found in 15,394 unique genes. There were 593 genes with significant differences in SNV number between HGSC samples and fallopian tube samples (FDR adjusted *p*-value < 0.05). Appendix A shows the 25 top genes with significant differences. The multivariate lasso regression analysis including all significant SNVs in the univariate analysis (*n* = 593), resulted in one gene, *CRIPS2*, with differences in SNVs that predicted EEC with an excellent performance, AUC of 1.00 (Figure 4). The presence of SNVs was protective of EEC (OR < 1).

In the EEC dataset, including 62 cases, there were a total of 658,391 variants: frame-shift deletion/insertion, in-frame deletion/insertion, missense, non-sense or non-stop mutations, and splice site alterations. Appendix A summarized these variations processed with *maftools*. The normal endometrial tissue set (with 36 controls) had 137,382 SNVs (Appendix A). All these variants were found in 15,792 unique genes. There were 2925 genes with significant differences in SNV number between EEC samples and normal endometrial samples (FDR adjusted *p*-value < 0.001). Appendix A shows the 40 top genes with significant differences, and the lasso regression analysis (Appendix A) where only differences in SNV number for the gene *PQBP1* were informative for prediction of EEC, protecting against the presence of cancer (OR = 0.81).

Some of the SNV with significant differences between HGSC and tubal samples and between EEC and normal endometrial samples were also associated with clinical outcomes, such as overall survival (HGSC in Figure 5 and EEC in Appendix A).

Location of gene variations that had significant SNV count and survival differences between the ovarian and endometrial datasets are represented with lollipop plots in Appendix A.

### 3.3. Correlations between Significant 16S rRNA Counts and SNVs

We correlated all significantly different 16S rRNA log2-transformed counts in the univariate analysis between HGSC and tubal samples (*n* = 13, FDR adjusted *p*-value < 0.05), with significant SNVs between the same samples (*n* = 593, also FDR-adjusted). Significant correlations were considered those with FDR adjusted *p*-value of <0.001 and were observed between 11 different 16S rRNA genes and 160 different genes harboring SNVs (Figure 6, Appendix A). Genera *Escherichia/Shigella* and *Pantoea* had most significant correlations with genes harboring SNVs.

Correlations between significant normalized, log2-transformed, 16S rRNA and SNV counts in the univariate analysis of EEC vs. normal endometrial samples resulted in many significant associations between bacterial and tumoral transcripts, even after adjusting for multiple comparisons (FDR-adjusted *p*-value < 10^−5^). There were 107 bacterial transcripts (out of 112 significant in the univariate analysis) and 948 genes (out of 2925) with significant correlations. Figure 7 depicts the most significant correlations: 15 16S rRNAs and 134 genes harboring SNVs (all significant correlations are represented in Appendix A). Genera *Rhodopseudomonas* and *Proteus* had the most significant correlations with these significant genes harboring SNVs.

### 3.4. Validation Analysis between Control and Cancer Samples in TCGA Dataset

In the TCGA HGSC database (*n* = 352) there were 868 taxa and 2,253,657 SNVs (Appendix A). Within this database we found the same 13 16S rRNA transcripts that were significantly different between HGSC and tubal samples in the UI dataset. In addition, we found 592 out of 593 genes with SNVs in the TCGA dataset that were significantly different between our HGSC and the fallopian tube dataset somatic genomic variation. In the UI dataset, significant correlations were observed between 11 different 16S rRNA transcripts and 160 genes harboring SNVs. In the TCGA HGSC dataset, significant correlations were observed between 10 out of the original 11 different 16S rRNA genes and 90 (out of the initial 160) genes harboring SNVs (Appendix A). Like the UI database, genera *Escherichia/Shigella*, *Nevskia*, *Methyloversatilis*, and *Ralstonia* also had the most significant correlations with genes harboring SNVs. Additionally, *Nevskia*, *Ralstonia*, and *Pantoea* validated in the TCGA HGSC database to have significant correlations with genes carrying SNVs, were part of the multivariate model predictive of HGSC.

In the TCGA EEC database (*n* = 408), there were 1565 taxa and 4,320,888 SNVs (Appendix A). Within this database we found 107 16S rRNA transcripts that were significantly different between EEC and normal endometrial samples in the UI dataset. In addition, we found 2831 out of 2925 genes with SNVs in the TCGA dataset that were significantly different between our EEC and normal endometrial somatic genomic variation. The UI EEC database had substantially more significant correlations between 16S rRNA transcripts and genes with SNVs than the UI HGSC. That was also true with the TCGA EEC database; significant correlations were observed between 79 different 16S rRNA transcripts (out of 107) and 447 genes harboring SNVs (out of 2831, with FDR-adjusted *p*-value < 10^−5^, Appendix A). The common bacteria genera between the TCGA and UI databases that correlated with EEC genomic variation were *Proteus*, *Anaerobacterium, Klebsiella*, *Rugamonas*, *Parabacteroides*, *Serratia*, *Erwinia*, *Flavobacterium*, and unclassified *Fusobacteriaceae*. Additionally, genera *Desulfomicrobium*, *Parabacteroides*, and *Proteus,* that were validated in the TCGA EEC database to have significant correlations with genes carrying SNVs, were also part of the multivariate model predictive of EEC.

### 3.5. Pathway Enrichment Analysis

Genes with SNVs that were significantly correlated to 16S rRNA transcripts in the HGSC database (*n* = 160) were introduced in an enrichment pathway analysis. Overrepresented and statistically significant pathways in this list of genes are displayed in Figure 8A. Most of these pathways are involved in ciliary physiology, inflammatory response, and signaling (Appendix A).

Likewise, we introduced all genes with SNVs that were significantly correlated to 16S rRNA transcripts in the EEC database (*n* = 948) in an enrichment pathway analysis, and again, most of these pathways were involved in neurodegenerative diseases, the novel Coronavirus disease—COVID-19 pathway, or internal protein processing (Figure 8B, Appendix A).

## 4. Discussion

Transcripts from microorganisms are found in RNAseq experiments of samples from normal tissues and cancers, including within the female upper genital tract [11,12,34]. Furthermore, microbial changes have been observed from the vagina to the peritoneal fluid [11]. Our present study confirms a changing landscape in microbial communities as we progress into the upper genital tract, with more diversity in lower levels of the tract. In addition, we observed significant differences between the normal tissue microbiome and microbial communities of neoplastic tissues, with more diverse significant bacteria genera in EEC than in HGSC when we compared to control samples. The determination of these microbial communities was performed through 16S rRNA sequencing and expression, a sensitive method that has become one of the gold standards of taxa identification of microbial communities without having to rely on growing organisms in pure culture [23]. It must be noted that, in this study, all samples were obtained during surgery, therefore they were collected under otherwise sterile conditions.

Cancers have characteristic patterns of mutations and genomic alterations. SNVs are some of the most common sources of genomic variation in cancer. SNVs can also be distinctive of each cancer, so much so that patterns of SNVs could be used to create models that would discriminate cancerous tissue from benign tissue [13]. In the present study, we characterized patterns of somatic SNVs, detected with the best recommended practices of genome sequencing [35], from HGSC and EEC. Some of those identified SNVs were also associated with clinical outcomes. Furthermore, we describe a multivariate lasso regression model that distinguishes HGSC and EEC with the determination of SNVs in a single gene with high performance (measured in AUC). Previous work from our laboratory has confirmed that high-performance models for detection of HGSC are possible with SNV analysis [13]. However, the overarching goal of this study was not to detect HGSC and EEC SNVs but to determine the association between this genomic variation and changes in microbial communities in neoplastic tissues.

Some of the genera most informative for HGSC in the lasso multivariate analysis and in the correlation with significant genomic SNVs, *Escherichia/Shigella*, *Pantoea*, and *Ralstonia*, have been associated with other types of cancer, including bladder, pancreatic, cholangiocarcinoma, colorectal, breast, and oral malignancies [42,43,44,45,46]. Likewise, variants of the dynein axonemal heavy chain (*DNAH*) family have been previously associated with breast and ovarian cancer [47]. The pathway enrichment analysis of those genes with significant SNVs between cancer and controls that were also associated with microbiological communities’ changes showed a predominance of pathways involved in internal signaling and immune response to infection, including to *E. coli*, one of the most correlated genera. Interestingly, while ciliary pathology was involved in the HGSC cohort, lipid metabolism seemed to be predominant in the EEC cohort.

In view of these results, we could postulate hypotheses on how the changing landscape of microbial communities may affect the environment for local cellular transformation. There are common patterns for both types of cancer, HGSC and EEC. The presence of an altered microbiome may facilitate the selection of cells with distinct genomic/genetic variation in the local epithelium, as demonstrated by significant correlations of different bacterial communities and different SNVs between cancer and control samples. These different genomic variants may also interact or modify the immune response of the normal epithelium against bacterial colonization (observed in the pathway analysis), with potential improvement in adaptation to the environment for those selected bacteria. At the same time, these variants may affect the normal cellular signaling pathways that modify the risk for malignant transformation. There are potential specific mechanisms of microbiome interaction for ovarian or endometrial cancers. Genes that affect ciliary physiology are overrepresented in the pathway analysis of genes with SNVs that are correlated with changed microbial communities in HGSC samples. Tubal transit is a complicated mechanism that includes interaction between muscle contractions, ciliary activity, and the flow of tubal secretions [48]. The role of ciliary motion in this process is key and may affect various pathological states, including sterility, endometriosis, and microbial infection. In the pathway analysis of genes with differential SNVs in the EEC database we observed a preponderance of pathways involving lipid metabolism. Estrogen association with EEC is well documented, especially with imbalanced estrogen production and signaling [49]. In vivo, estrogen is synthesized from cholesterol which is an intrinsic component of lipid metabolism [50]. Unbalances of estrogen production has been associated to a wide range of tumors, besides endometrial and breast cancers, and the microbiome has been implicated in some of these disruptions [50]. A changing microbiome that modulates its own microenvironment for its progression, may also result in epithelial cell selection with genomic variation that predisposes to malignant transformation [51].

*DNAH11* encodes a ciliary outer dynein arm protein, a member of the dynein heavy chain (*DNAH*) family and was associated with one of the main significant pathways, ciliopathies (WP4803). This gene is a microtubule-dependent motor ATPase involved in the movement of respiratory cilia. Variations in this gene have been implicated in causing Kartagener Syndrome and Primary Ciliary Dyskinesia (PCD) [52]. As we noted before, members of the *DNAH* family have been previously linked with breast and ovarian cancer [47]. Components of the VEGFA-VEGFR2 signaling pathway (WP3888), such as *IGFBP3*, also have been involved in the cholesterol metabolism pathway, with potential effects on several aspects of the proposed pathophysiology between altered bacterial communities and genomic variation in upper genital tract tumors. *AKT2* is involved in some of the most common signaling pathways, belonging to a subfamily of serine/threonine kinases. The gene serves as an oncogene and is believed to contribute to the malignant phenotype of some pancreatic cancers. *AKT2* is one of the components of the PI3K-Akt signaling and, with the JAK/STAT3 signaling pathway, has a critical role in promoting malignant transformation in human tumors [53]. The JAK/STAT3 pathway is also activated and associated with tumor progression and poor prognosis in patients with ovarian cancer [54], and is a potential target for new therapeutic agents [55]. Critically ill patients with SARS-CoV2 viral infections also seem to hyperactivate the JAK1/2-STAT1 signaling pathway [56]. Pathway enrichment analyses demonstrated how changes in microbial communities may affect intracellular signaling pathways due to their correlation with variation-altered genes. However, to understand the mechanism of interaction of bacteria with the underlying epithelium and the signaling pathways that may be affected, such as JAK/STAT3 or PI3K-AKT/mTOR, functional analyses are needed. Furthermore, functional analyses are better suited to assess whether any changes in the microbiome could help in selecting targeted therapies against the JAK1/STAT3 signaling pathway such as JAK inhibitors [54], or mTOR inhibitors to target the PI3K-AKT/mTOR pathway [55].

While our findings are novel and provide unique insights, a limitation of this study is the retrospective nature of the design. The RNAseq experiments and clinical data collection were not originally designed to be functional analyses of changing microbial communities. Thus, despite robust statistical significance in the association with cancer and correlation with genomic variation, these results must be examined and validated mechanistically to establish causality. Moreover, to find appropriate controls for the EEC cases, we downloaded RNAseq data from publicly available, reliable sources (GEO). Neither did we use environmental swabs to remove background noise/contamination from the samples. Nevertheless, the specimens were not collected under the same conditions as in all cases. Because RNAseq measures the count of transcripts, we considered it as an objective measurement, subjected to little or no bias. RNAseq results were processed and analyzed identically in both cases and controls, including normalization and log2-transformation. A major strength of this study is that this is a large genomic dataset with clinical outcomes, collected at a single tertiary medical center, which ensured protocol consistency in sample collection and analysis procedures. This allowed for statistical correlation between genomic data within this population. Another major strength of this study is the validation of initial results in the TCGA datasets for HGSC and EEC. TCGA is a well-known independent, high-quality, publicly available database of genomic and clinical data. We found similar differential 16S rRNA transcripts from both databases in TCGA validation, and similar correlations were found with significant SNVs. The similarity between the initial analysis and validation made our results plausible. The 16S rRNA detection for taxonomic composition of microbial communities is a well-established methodology, and the identification of the microbiome in the upper genital tract is no longer suspected to be contamination [12]. Therefore, it is very unlikely that the results were biased due to contamination. Validation of RNAseq results in an independent dataset along with independent lab processing also minimizes the possibility of bias due to contamination.

Our findings provide ground for speculation about potential avenues for research and cancer prevention. First, to determine what is normal, a female genital tract microbiome mapping is critical. Then, we could theoretically study the effect of certain agents, such as talk [57], antibiotics, and immunosuppressants administered locally and systemically, nutritional patterns and status, and the effect of other factors such as complex diseases (diabetes), obesity and autoimmune disorders. The American Cancer Society (ACS) already recommends to perform routine gynecologic exam to detect alterations in the micro-environment of the cervix for cancer screening: individuals with a cervix should initiate screening at age 25 years and undergo primary human papillomavirus (HPV) testing every 5 years [58]. Epidemiological studies could piggy-back on these recommendations and monitor the health status of the genital microbiome.

## 5. Conclusions

We have identified different bacterial communities and different genomic variation (SNV) between gynecological cancers and normal controls. Some of these genomics features strongly correlated with each other. The pathway analysis resulting from the correlated SNV genes established a pattern that may explain how changing bacterial landscapes could lead to these cancers. Some of the components of these pathways could also identify potential candidates for targeted therapy in these tumors. However, what is highly thought provoking is the possibility that these studies may contribute to cancer prevention in women at risk by detecting altered bacterial communities in the upper genital tract epithelium.

## Figures and Tables

**Figure 1 cancers-15-03316-f001:**
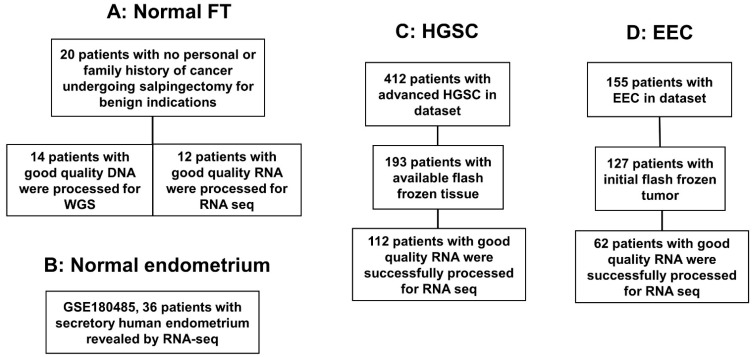
Flowchart of patients included in the analysis. (**A**) Normal fallopian tubes (FT) samples from patients with no risk factors and no personal or familiar history of ovarian cancer. Out of the 20 samples, 14 had DNA for whole genome sequencing (WGS) and 12 for RNA sequencing (RNAseq). (**B**) A total of 36 samples from healthy patients who were evaluated in a fertility study underwent RNAseq, (GEO ID# GSE180485). (**C**) Samples from high-grade serous ovarian cancer (HGSC) patients that underwent surgery at the University of Iowa (UI) and had their tumors sequenced (RNAseq), *n* = 112. (**D**) Samples from endometrioid endometrial cancer (EEC) patients that underwent surgery at the University of Iowa (UI) and had their tumors sequenced (RNAseq), *n* = 62.

**Figure 2 cancers-15-03316-f002:**
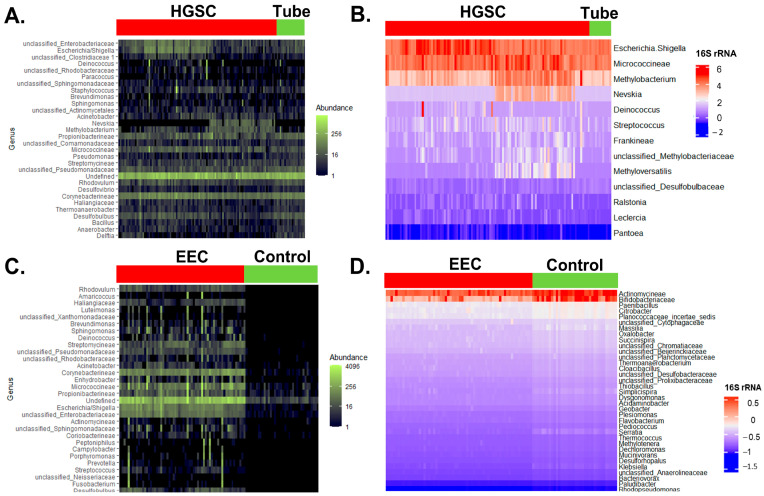
Comparison of 16S rRNA gene expression between control and cancer samples. (**A**) Heatmap of the normalized 30 most frequent bacterial 16S rRNA counts found by RNAseq and represented by condition: normal fallopian tube vs. HGSC. Abundance of counts is represented in the green scale (analysis performed with *phyloseq*). (**B**) Heatmap of the 16S rRNA log2-transformed, normalized counts found by RNAseq between a normal fallopian tube and HGSC that were different in the univariate logistic regression analysis, *n* = 13 (FDR adjusted *p*-value < 0.05). Log2 counts are represented in the blue-white-red scale (analysis performed with *DESeq2*). (**C**) Heatmap of the normalized 30 most frequent bacterial 16S rRNA counts found by RNAseq and represented by condition: normal endometrium vs. EEC. Abundance of counts is represented in the green scale (*phyloseq*). (**D**) Heatmap of the 35 top 16S rRNA log2-transformed, normalized counts (out of 112) found by RNAseq between normal endometrium and EEC that were most different in the univariate logistic regression analysis (FDR adjusted for multiple comparisons). Log2 counts are represented in the blue-white-red scale (*DESeq2*).

**Figure 3 cancers-15-03316-f003:**
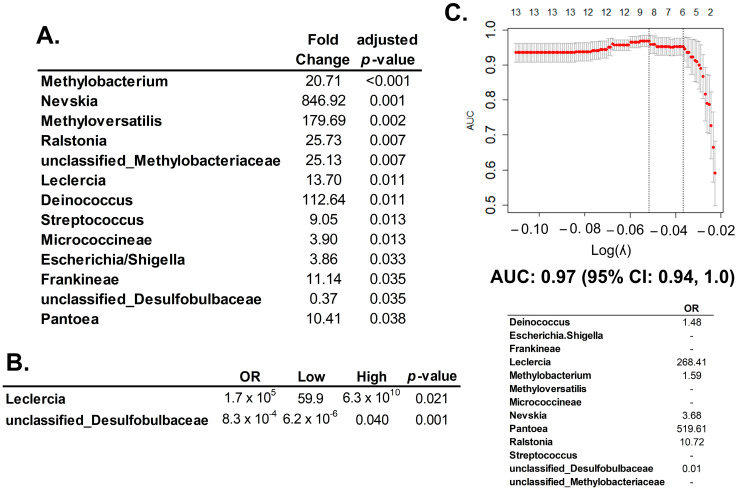
Differential gene expression of 16S rRNA betwen HGSC and normal fallopian tube. (**A**) Significant 16S rRNA expression between a normal fallopian tube and HGSC specimens (*n* = 13) calculated by multiple univariate logistic regression analyses (and adjusted for multiple comparisons with BH method), using the *DESeq2* package. Twelve of the species identified by 16S rRNA expression were associated with a higher risk for HGSC (OR > 1), and only one of them was protective against HGSC, which was unclassified *Desulfobulbaceae*. (**B**) Multivariate logistic regression that included all significant 16S rRNA expressions from the univariate analysis (Panel A). Only the expressions of two bacteria were independently significant in this analysis: genera *Leclercia* and unclassified *Desulfobulbaceae*. (**C**) The lasso multivariate regression model included all 16S rRNA counts significant in the univariate analysis (*n* = 13). In the upper panel: Graphic representation of the multivariate lasso analysis: number of variables are in the upper axis; left axis is the area under the curve (AUC) measurement; lower axis represents the value of lambda: used as tunning parameter to optimize the model. The optimized AUC was 0.97 (95% CI: 0.94, 1.0), between the dotted lines. In the lower panel: The lasso multivariate regression model: out of the 13 significant counts in the univariate analysis, seven 16S sRNAs remained informative for prediction HGSC. Six of them increased risk (OR > 1) and one protected from HGSC (OR < 1). Graphics were generated with R package *glmnet*.

**Figure 4 cancers-15-03316-f004:**
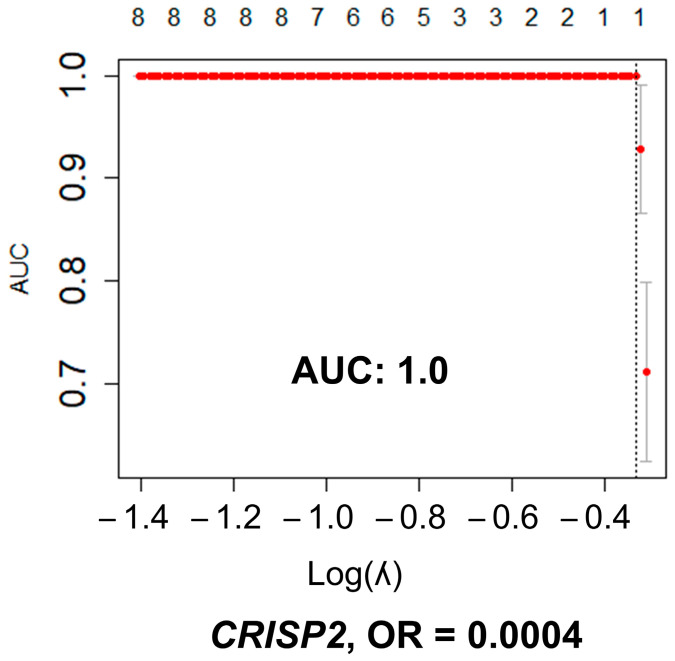
Multivariate lasso regression model between EEC and control endometrial samples. The lasso multivariate regression model included all SNVs significant in the univariate analysis (*n* = 593). Graphic representation of the multivariate lasso analysis: number of variables are in the upper axis; left axis is the area under the curve (AUC) measurement; lower axis represents the value of lambda: used as tunning parameter to optimize the model. The optimized AUC was 1.0, between the dotted lines. In the model, only differences in SNV number in *CRISP2* remained informative for the prediction of HGSC, protecting against the presence of cancer (OR < 1). Graphics were generated with R package *glmnet*.

**Figure 5 cancers-15-03316-f005:**
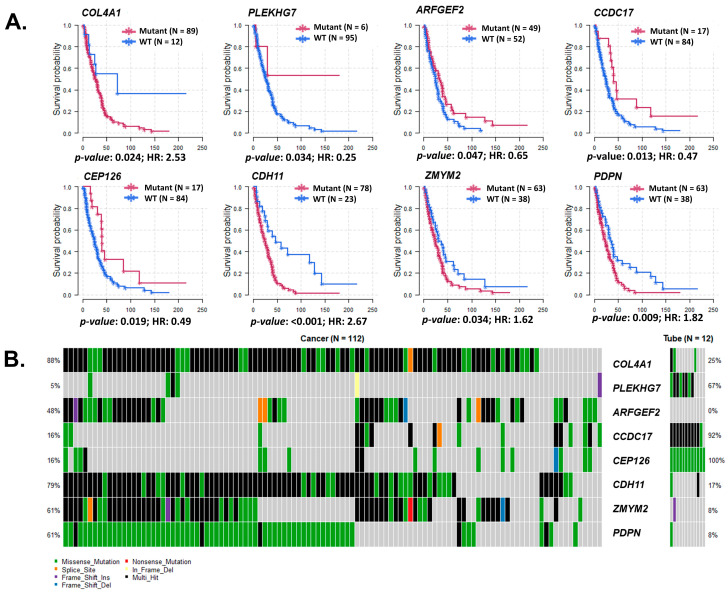
Survival analysis of genes that had significant SNV differences between HGSC and normal fallopian tube samples. (**A**) Survival curves with respective hazard ratios (HR) of survival between genes with differences between wild type (WT) and genes with variations (Mut). Genes: COL4A1, PLEKHG7, ARFGEF2, CCDC17, CEP126, CDH11, ZMYM2, and PDPN. (**B**) Variant classification and number of SNV of these significant genes with different survivals. SNV classification: Frame-Shift deletion/insertion, In-Frame deletion/insertion, missense, non-sense or non-stop mutations, splice site, translation start site, and multi-hit.

**Figure 6 cancers-15-03316-f006:**
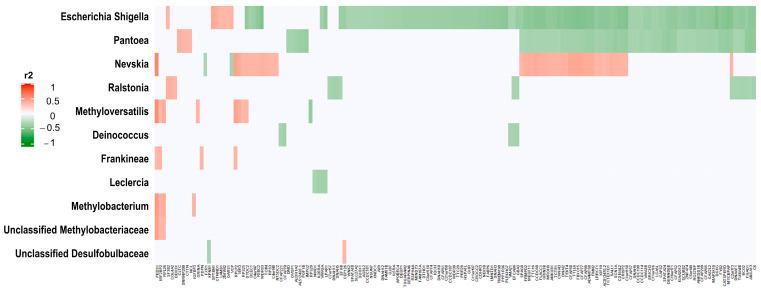
Correlations of HGSC 16S rRNA counts with HGSC SNV counts. The graphic represents the correlation (measured in r^2^) between significantly different 16S rRNA expression between HGSC and tubal samples (*n* = 13, FDR-adjusted *p*-value < 0.05), and significantly SNV counts between the same samples (*n* = 593, also FDR-adjusted). The correlation was performed with log2 transformed and normalized counts (SNV and 16S rRNA). Significant correlations were observed between 11 different 16S rRNA genes (y axis) and 160 different genes harboring SNVs (x axis) (FDR-adjusted *p*-value of <0.001). For further details see Appendix A. Red represents significant direct correlations: more 16S rRNA counts, more SNV gene counts. Green represents significant inverse correlations: more 16S rRNA counts, less SNV counts.

**Figure 7 cancers-15-03316-f007:**
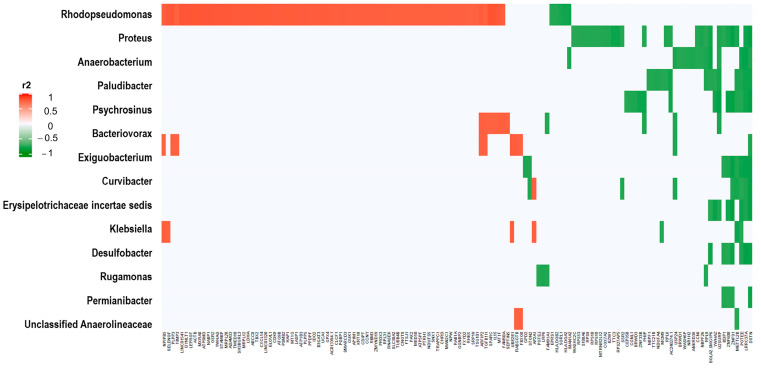
Correlations of EEC 16S rRNA counts with EEC SNV counts. The graphic represents significant correlations (measured in r2) between the 15 most significantly different 16S rRNA normalized transcripts and the 134 significant gene-harboring SNVs from the univariate analysis between EEC and normal endometrial samples (FDR adjusted *p*-value < 10^−5^). For further details see Appendix A. Red represents significant direct correlations: more 16S rRNA counts, more gene SNVs. Green represents significant inverse correlations: more 16S rRNA counts, less SNVs.

**Figure 8 cancers-15-03316-f008:**
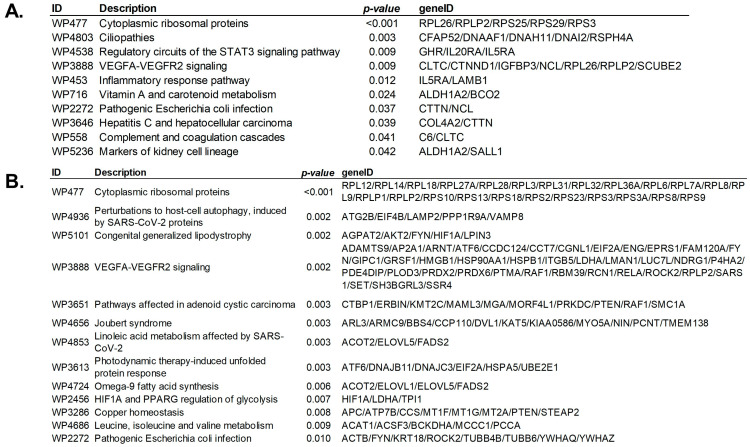
Enrichment pathway analysis of genes with SNVs that were significantly correlated to 16S rRNA transcripts in HGSC and EEC. (**A**) Significantly enriched *WikiPathways* for the genes with differential SNVs in the HGSC database, with predominance of ciliary physiology, immune response to infection, including to *E. coli*, and signaling pathways. (**B**) Significantly enriched *WikiPathways* for the genes with differential SNVs in the EEC database, also with predominance of immune response to infection, including to *E. coli*, lipid metabolism, and signaling pathways. ID: *WikiPathways* ID (WP and number); *p*-value: significant FDR-adjusted *p*-values < 0.05; geneID: symbols of genes correlated with 16S rRNA transcripts that are located in specified pathways.

## Data Availability

Clinical data is not publicly available due to patient privacy. Datasets with RNA-seq can be browsed by their accession number: GSE156699. The validation part of this study was performed in silico, with de-identified publicly available data. All data from TCGA is available at their website: https://portal.gdc.cancer.gov/, accessed on 20 January 2020. Software utilized by this study is also publicly available at Bioconductor website: http://bioconductor.org/, accessed on 20 January 2020. Software and transcripts used for the study can be found at the VITCOMIC2 GitHub site: https://github.com/h-mori/vitcomic2 (accessed on 2 March 2023).

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
