# Peer review of "Microbial Communities in Gynecological Cancers and Their Association with Tumor Somatic Variation"

_cancers, 2023, doi:10.3390/cancers15133316_

Round 1
Reviewer 1 Report
First of all, congratulations on the very nice work.
But I would like to ask the following:
1- Regarding the analysis method for 16S rRNA-sequencing, you said you chose to use this VITCOMIC pipeline. Can you please justify the choice? This pipeline has only 18 citations while other methods, such as qiime2, have hundreds and are standard practice for the 16S analysis.
2- Can you add the version of the RDP database used? There are frequent updates, so mentioning the version is necessary.
3- Could you justify the selection of a 94% identity when analysing the 16S? As far as I am aware, the standard is in the range of 98-99%.
4- Also, regarding the 16s rRNA-sequencing, you call it an RNA-seq method, but here one uses DNA, not RNA. The Illumina 16S Metagenomic Sequencing Library Preparation protocol uses DNA as input, and the PCR primers target the variable regions of the 16S DNA gene for the amplicon PCR. Or did I misunderstand what you meant?
5- Regarding the reproducibility of your finds, I would highly appreciate adding a github link with the analysis performed. I am talking about a jupyter notebook or R-markdown of the analysis for this request.
Finally, it is known that although the surgery environment is considered sterile when working with the microbiome, controls are needed for trace amounts of bacterial DNA, even in this environment. For example, the studies related to the placenta microbiome where trace amounts of a microbial signal are found on the samples although the placenta is considered sterile. So I would like to ask if you also had a control sample using environmental swabs to remove background noise/contamination from the samples.
Author Response
1- Regarding the analysis method for 16S rRNA-sequencing, you said you chose to use this VITCOMIC pipeline. Can you please justify the choice? This pipeline has only 18 citations while other methods, such as qiime2, have hundreds and are standard practice for the 16S analysis.
The reviewer is absolutely right, there are a wide variety of options to choose from for 16S rRNA analysis. We chose VITCOMIC2 because it suited well the type of analysis we wanted to perform. First, the second version of this pipeline had improved in detection accuracy, was faster in searching for sequence identity, easy to install and use, and easy to deploy in a workstation. On top of that, we did not require a lot of other utilities that other pipelines offered, like de-multiplexing or visualization. However, the main reason we use this pipeline was because of the flexibility of use. VITCOMIC2 was divided into several steps that could be applied to the data we had and would benefit our study goals. One of the main objectives of the study was the association of bacterial communities with the gynecologic cancer phenotype. To assess this association, we wanted to evaluate not only which bacteria were present but those bacteria that were transcriptional active. We believe that those bacteria that are more transcriptionally active are going to have a higher impact in the neighboring epithelium. After the initial step of QC, VITCOMIC2 uses SeqKit (https://github.com/shenwei356/seqkit/), a tool for FASTA/Q manipulation. This tool was interesting to us because would convert both DNA and RNA into FASTA format. Then, the tool MAPseq (https://github.com/jfmrod/MAPseq) would do the classification (alignment) with the FASTA file using the ref file as template. At this point MAPseq could use FASTA files either from converted DNA or RNA FASTQ files. We used it for both. There are some reports that show MAPseq had comparable, if not superior performance than QIIME2 (Almeida, A., et al. Benchmarking taxonomic assignments based on 16S rRNA gene profiling of the microbiota from commonly sampled environments. Gigascience 2018, 7, doi:10.1093/gigascience/giy054.)(new reference added).
We reviewed methods to clarify this point.
2- Can you add the version of the RDP database used? There are frequent updates, so mentioning the version is necessary.
The initial RDP database used by VITCOMIC2 authors’ was RDP, release 11, update 2. They performed some modifications of the database to finally obtain a curated database with 28,977 high-quality 16S rRNA gene sequences (The list if in Ref #22, Additional file 2 and Additional file 3).
We will add this information to methods.
3- Could you justify the selection of a 94% identity when analysing the 16S? As far as I am aware, the standard is in the range of 98-99%.
We used the standard settings recommended by VITCOMIC2 authors in REF# 22: ‘…identity ≥94%...’ (page 52). We only change the setting for the alignment length threshold. When using endometrial TCGA, sequencing for some batches was performed with older technology resulting in transcripts around 50 bp.
Added to the manuscript.
4- Also, regarding the 16s rRNA-sequencing, you call it an RNA-seq method, but here one uses DNA, not RNA. The Illumina 16S Metagenomic Sequencing Library Preparation protocol uses DNA as input, and the PCR primers target the variable regions of the 16S DNA gene for the amplicon PCR. Or did I misunderstand what you meant?
We are sorry for the confusion. This is a retrospective study that used already performed RNA and DNA sequencing products. We performed a whole DNA exome sequencing and total RNA sequencing from gynecological cases and controls, and libraries for both were built at that time. The selection of the 16S amplicons and/or transcripts were performed during the alignment with the reference sequence, at the MAPseq step.
5- Regarding the reproducibility of your finds, I would highly appreciate adding a github link with the analysis performed. I am talking about a jupyter notebook or R-markdown of the analysis for this request.
We used the scripts as recommended by VITCOMIC2 authors that can be found at https://github.com/h-mori/vitcomic2. The only modification was decreasing the cut off for the alignment length, because of older TCGA batches, as stated before. The modification was in the ‘PileupGenerMDB’ perl file.
We will add this GitHub link to the manuscript.
Finally, it is known that although the surgery environment is considered sterile when working with the microbiome, controls are needed for trace amounts of bacterial DNA, even in this environment. For example, the studies related to the placenta microbiome where trace amounts of a microbial signal are found on the samples although the placenta is considered sterile. So I would like to ask if you also had a control sample using environmental swabs to remove background noise/contamination from the samples.
Sorry, we did not. This is a retrospective study. We did not design the study for these specific conditions. The retrospective nature of the study is one of its weaknesses.
We will add this comment to the limitations paragraph.
Reviewer 2 Report
The MS: ‘Microbial communities in gynecological cancers and their association with tumor somatic variation’ is interesting and valuable since it highlights a future possibility of the prevention of some aggressive gynecological cancers (e.g., high-grade serous ovarian cancer (HGSC) and endometrioid endometrial cancer (EEC)) especially in women at risk, via detecting the altered bacterial communities in the upper genital tract epithelium.
The Authors may consider some suggestions for their revision.
In the Discussion [p # 13, 14], the authors could briefly explain the term: ‘lipid metabolism’
in the sentence: ‘Interestingly, while ciliary pathology was involved in the HGSC cohort, lipid metabolism seemed to be predominant in the EEC cohort’.
in the paragraph: ‘In the pathway analysis of genes with differential SNVs in the EEC database, we observed a preponderance of lipid metabolism pathways.
Does ‘lipid metabolism’ is relevant primarily to abnormal lipid metabolism (e.g., dyslipidemia – what kind?) in the context of imbalanced estrogen production and signaling?
Also, in the Discussion, the authors might comment on the potential options for medical staff and patient education avenues, which can promote awareness of the fact that changing bacterial landscapes (e.g., through nutrition, antibiotics, lifestyle, etc.) may be related to gynecological cancers, their behavior, and patient screening, diagnostic/therapeutic management, and clinical outcomes. This, in turn, may contribute to a more precise detection of potential candidates for the targeted anticancer therapies (who would be well-educated and involved in their treatments).
Author Response
The MS: ‘Microbial communities in gynecological cancers and their association with tumor somatic variation’ is interesting and valuable since it highlights a future possibility of the prevention of some aggressive gynecological cancers (e.g., high-grade serous ovarian cancer (HGSC) and endometrioid endometrial cancer (EEC)) especially in women at risk, via detecting the altered bacterial communities in the upper genital tract epithelium.
The Authors may consider some suggestions for their revision.
- In the Discussion [p # 13, 14], the authors could briefly explain the term: ‘lipid metabolism’
- in the sentence: ‘Interestingly, while ciliary pathology was involved in the HGSC cohort, lipid metabolism seemed to be predominant in the EEC cohort’.
- in the paragraph: ‘In the pathway analysis of genes with differential SNVs in the EEC database, we observed a preponderance of lipid metabolism pathways.
Does ‘lipid metabolism’ is relevant primarily to abnormal lipid metabolism (e.g., dyslipidemia – what kind?) in the context of imbalanced estrogen production and signaling?
Very good question. We hypothesize that is more related to the imbalanced estrogen production and signaling, which has been associated to Type I endometrial cancer genesis and progression.
This point was added to the discussion.
- Also, in the Discussion, the authors might comment on the potential options for medical staff and patient education avenues, which can promote awareness of the fact that changing bacterial landscapes (e.g., through nutrition, antibiotics, lifestyle, etc.) may be related to gynecological cancers, their behavior, and patient screening, diagnostic/therapeutic management, and clinical outcomes. This, in turn, may contribute to a more precise detection of potential candidates for the targeted anticancer therapies (who would be well-educated and involved in their treatments).
A new paragraph at the end of the discussion has been added to comment on these issues.
Round 2
Reviewer 1 Report
Thank you for all clarifications.
I appreciate that you took into account my comments.